# Small-scale adversarial perturbations expose differences between predictive encoding models of human fMRI responses

## Abstract

Artificial neural network-based vision encoding models have made significant strides in predicting neural responses and providing insights into visual cognition. However, progress appears to be slowing, with many encoding models achieving similar levels of accuracy in predicting brain activity. In this study, we show that encoding models of human fMRI responses are highly vulnerable to small-scale adversarial attacks, revealing differences not captured using predictive accuracy alone. We then test adversarial sensitivity as a complementary evaluation measure and show that it offers a more effective way to distinguish between highly predictive encoding models. While explicit adversarial training can increase robustness of encoding models, we find that it comes at the cost of brain prediction accuracy. Our preliminary findings also indicate that the choice of model features-to-brain mapping might play a role in optimizing both robustness and accuracy, with sparse mappings typically resulting in more robust encoding models of neural activity. These findings reveal key vulnerabilities of current models, introduce a novel evaluation procedure, and offer a path toward improving the balance between robustness and predictive accuracy for future encoding models.

## 1   Introduction

Artificial neural networks (ANNs), loosely inspired by the architecture of the visual cortex, have become the leading models for understanding human vision [1–3]. These models excel not only at complex tasks like object recognition (e.g., ImageNet classification) but also provide a valuable framework for studying visual cognition more broadly [4–6]. ANN-based encoding models, which map neural network features to brain activity, have unlocked a key ability to predict responses at the level of single neurons [7], voxels [8], entire brain regions [9, 10], and even human and non-human primate behavior [11–13]. Early work established a link between a model's performance on complex tasks (like ImageNet) and the ability to predict brain responses: better task performance typically translated to better brain/behavioral predictions [14, 1, 15]. However, this relationship has plateaued; despite continual improvements in task performance, gains in brain prediction accuracy (henceforth predictivity) have largely stalled. This observation raises critical questions: Are models with similar predictivity learning the same features, or are key differences going unnoticed? Is there a more effective metric that can reveal these differences and help us identify the better models, even when their predictivity appears to be equally high? In this work, we show that small, imperceptible (to humans) adversarial attacks on predictive encoding models can reveal meaningful differences, providing a sharper lens to evaluate their fidelity as models of the brain.

The concern that encoding model predictivity has plateaued is not new [14, 9, 10, 15, 16]. This stagnation has sparked two major responses within the field. On one front, this challenge has driven

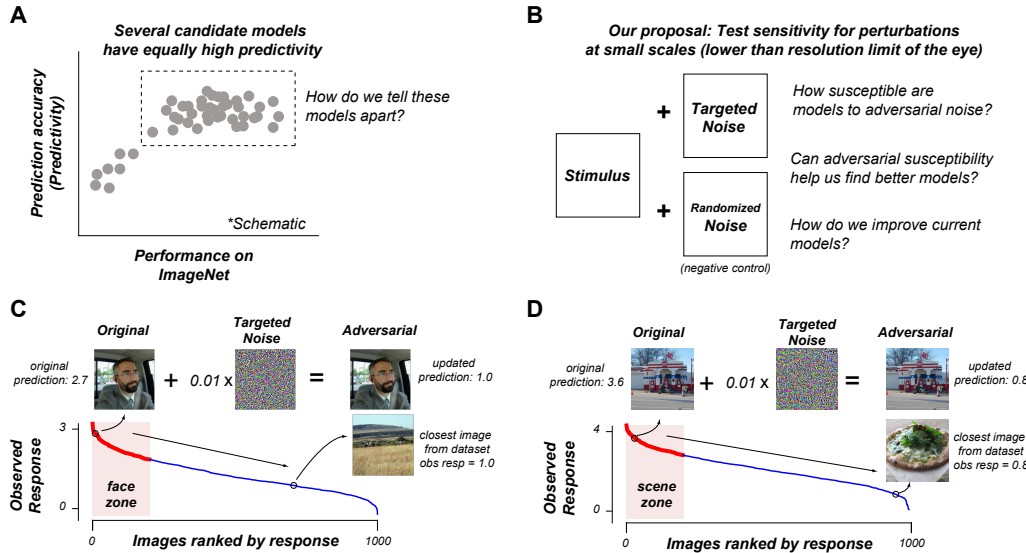

Figure 1: **Motivation, central questions and example adversarial perturbations on encoding models. A.** Schematic illustrating the trend observed in previous studies: many encoding models show similarly high predictions on brain data. Performance on ImageNet is shown on the x-axis, and prediction accuracy on brain data is shown on the y-axis. A similar figure with actual data can be found in previous work [1]. **B.** Strategy and central questions. For a given stimulus, we generate a targeted noise pattern (and a randomized control) to assess how sensitive the encoding model is to adversarial noise. **C.** Example of an adversarial perturbation applied to the FFA encoding model (VGG16). The model's prediction for the original face stimulus (top left) is significantly altered when a targeted imperceptible noise pattern is added (top middle). The modified image produces a much lower response (top right), similar to that for a non-preferred stimulus (bottom). The x-axis shows images sorted by response, and the y-axis represents the FFA's response. The red region highlights the response for the preferred category (faces). **D.** Same as C, but for an example scene stimulus and the PPA.

the development of *entirely new models* incorporating aspects of brain-like operations (like recurrence [17, 18, 2, 19, 20]) or by directly aligning with behavioral or neural data [21–23]. The second front challenges predictivity as the primary metric altogether, advocating for *alternative evaluation methods* like centered kernel alignment [24–27] or single-neuron selectivity [28, 29] to capture more nuanced aspects of brain-model alignment. In this work, we are advocating a slightly different strategy. Predictivity must remain a vital benchmark measure of our models: predictive models have enabled new understanding of brain function including the ability to modulate responses in the visual cortex [30–33]. However, predictivity alone is insufficient, especially when we are limited by data. We propose complementing predictivity measures with additional evaluation metrics. Here we introduce adversarial sensitivity as a potential tool for stronger model evaluations.

Adversarial perturbations have long plagued AI systems. Previous work has shown that tiny, imperceptible changes to an image can drastically alter model predictions [34–40]. This issue has driven extensive research into making AI models more robust, particularly for mission-critical applications. Yet the impact of adversarial perturbations has received surprisingly little attention in vision neuroscience. Some work has explored "robustified" encoding models, either through training directly on neural data [41] to estimate neural robustness or by employing explicit robust pre-training to modify percepts [42–44]. To our knowledge, no study has directly examined the vulnerability of encoding models to targeted adversarial perturbations, the relationship between adversarial sensitivity and predictivity, or the impact of model mapping choices on the model's adversarial robustness. Understanding the bounds of our encoding models is crucial for progress. If imperceptible changes can distort model predictions, it raises concerns about their reliability in capturing true neural processes and ability to generalize to unseen data.

The central contribution of our work is threefold: (A) we demonstrate that encoding models are susceptible to small-scale adversarial attacks (Figures 1, 3), (B) we show that adversarial sensitivity is a potentially more effective way to differentiate between encoding models than predictive accuracy alone (Figure 3), and (C) we find that the choice of feature-to-brain mapping in encoding model can impact adversarial sensitivity, with sparse mappings producing relatively more robust models of neural activity (Figure 5).

## 2    Methods

### 2.1    fMRI Dataset

We used publicly available 7T fMRI data from the Natural Scenes Dataset (NSD) [45] for all analyses in this study. Specifically, we focused on the responses to 515 shared stimuli obtained from fMRI scans of eight subjects in category-selective brain regions. Each subject viewed these images three times over multiple experimental sessions. All analyses were conducted using version 3 of the dataset (betas_fithrf_GLMdenoise_RR), obtained directly from the NSD website. In this work we focused on the category-selective areas: fusiform face area (FFA) [46], extrastriate body area (EBA) [47], parahippocampal place area (PPA) [48], and the visual word form area (VWFA) [49]. To ensure the inclusion of only the most category-selective voxels, we applied a stringent threshold of $tval > 7$ for all analyses. Models were trained to predict the voxel and trial-averaged responses across subjects, as in previous work [9].

### 2.2    Encoding Model

Typical ANN-based encoding models consist of two components: embeddings from a specific layer of the artificial neural network (serving as the representational basis) and a trainable mapping function. This mapping is typically done through regularized linear regression, which projects the features into the response subspace of neural activity.

Formally, we input each of our training images (see cross-validation schema next) into a representational encoder $f$ and extract the latent feature vector $z_l \in \mathbb{R}^{C_l \times H_l \times W_l}$. These features are then passed through our mapping function $g : \mathbb{R}^{C_l \times H_l \times W_l} \to \mathbb{R}^n$, where $n$ is the dimensionality of the predicted neural data. To build the encoder model, we freeze $f$ (the weights of the representational encoder) and train the mapping $g$.

**Model architectures**: We considered eight pre-trained artificial neural network architectures that have been previously validated against brain data. These include ResNet-50 [50], VGG16 [51], Inception v3 [52], SqueezeNet v1 [53] , AlexNet [54], CORnet-RT [55], DenseNet [56], and MobileNet-v2 [57].

To investigate whether increasing robustness improves the prediction accuracy of the encoding models, we also used publicly available models that were robustified through adversarial training [58]. These models share the same architecture (ResNet-50) and learning rule but differ in the degree to which they are trained adversarially. More details on the robust models and their training can be found in [59].

**Encoding model mapping procedures:** In this study, we experiment with five different mapping functions: ordinary least squares regression (OLS), lasso regression, ridge regression, a two-layer multi-layer perceptron (MLP), and a convolutional neural network (CNN). The first three mapping functions generate direct brain predictions, while the latter two involve learning at least one additional layer of features. These new features may enhance the model's ability to predict brain responses and could provide more representational robustness. However, the regression methods are computationally faster and do not require extensive hyperparameter tuning for convergence. Our two-layer MLP and CNN both include one hidden layer with 128 units. Note that we used OLS mapping for the first half of the paper because it is the most computationally efficient and does not rely on any assumptions.

**Encoding model cross-validation procedure:** We used the 515 shared images across all 8 subjects from the NSD dataset. We trained the model on a randomly chosen set of 400 images and all results in the study are based on predicted responses based on the held-out 115 images.

In Section 3.5, we investigate the effect of $L_1$ readout regularization on the adversarial robustness of the encoding model. We fit each model to the data using only one randomly chosen subject (subj2),

testing six different values of the regularization coefficient $\alpha$ (0.0001, 0.001, 0.005, 0.01, 0.05, 0.1). The $\alpha$ value that maximized predictive accuracy for this subject was selected for further analysis. Importantly, all model evaluations were conducted using an independent metric (adversarial sensitivity) and across all subjects.

## 2.3  Evaluating encoding model robustness

We evaluated adversarial robustness against the Fast Gradient Sign Method (FGSM) [35]. FGSM attacks are bounded by the $L_\infty$ norm. That is, we find the maximum change $\delta$ (bounded by a "*perturbation budget*" $\epsilon$) predicted to change the response of a given voxel. A successful attack would drastically (and unrealistically) change the predicted response of the encoding model. We quantified the adversarial sensitivity $s_i$ for a given voxel using the method described in [41]. Specifically, we use a sensitivity metric $s_i$ defined as:

$$s_i = \max_{||\delta|| \leq \epsilon} \left( g(f((x))) - g(f((x+\delta))) \right)$$

There are two things to note about this metric. First, since $s_i$ is a measure of model *sensitivity*, high values on this metric would indicate lower adversarial robustness. We indicate this in several of our plots. The second is that since the metric does not have an upper bound, the results must not be interpreted across regions. While other forms of adversarial attacks exist in the literature, we focus on FGSM for simplicity and consistency.

## 2.4  Encoding Model Discriminability

We evaluate the ability of both metrics – adversarial robustness and model predictivity – to discrimininate encoding models of the brain. For each of the eight models evaluated, we compute the average sensitivity across all subjects and brain regions. We explore whether the spread of the adversarial robustness distribution of the encoding models will be greater than the spread of the model predictivity distribution (i.e., "adversarial robustness" serves as a better discriminative tool). To evaluate this, we test the variance and sparseness of both adversarial sensitivity and predictivity.

**Normalized Variance:** Since the scale of "sensitivity" (unbounded) and "predictivity" (bounded $-1$ to 1) are different, we cannot directly compare the variances. Instead, we first divide all accuracy and sensitivity values by their respective maximum value before reporting the variances (hence normalized variance).

**Sparseness:** We use the sparseness metric defined in [60, 61]. Specifically, for a distribution of values $P(r)$, sparseness (S) is computed with the following:

$$S = 1 - \frac{E[r]^2}{E[r^2]},$$

where $E[\cdot]$ denotes the expectation operator.

## 3  Results

Our investigation focuses on category-selective regions—specifically face, body, scene, and word-selective areas (FFA, EBA, PPA, and VWFA, respectively) from the Natural Scenes Dataset (NSD). These regions were chosen because of the extensive work on developing encoding models for them and because they provide the necessary foundational intuition for interpreting changes due to adversarial perturbations (Figure 1C, 1D). We specifically focus on *very small image perturbations* ($\epsilon \leq 3/255$) lower than the resolution limit of the human eye and hence imperceptible to humans. This is because the response of brain voxels to these targeted noise patterns remains currently unknown. Restricting our analysis to small magnitudes ensures that the adversarial sensitivities we detect are real and meaningful.

### 3.1  Several ANN-based encoding models predict voxel responses equally well

We first set out to replicate the previous finding that encoding models exhibit similar accuracy in predicting brain responses. To do this, we examined eight pre-trained neural network architectures

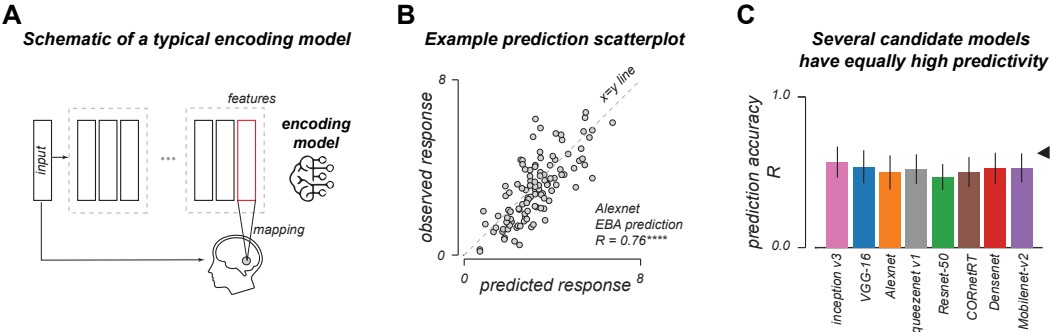

Figure 2: **Several encoding models have equally high predicitivity on fMRI data A.** Schematic outlining the construction of a typical encoding model. Features from an intermediate model layer (shown in red) are used to build a linear mapping function (indicated as "mapping") to predict responses in specific brain regions. **B.** Example scatterplot showing predicted (x-axis) versus observed responses for AlexNet in the EBA. The dotted line represents the x = y line, and each dot corresponds to a stimulus that was not used in model training (cross-validated). **C.** Bar plot showing various candidate encoding model architectures (x-axis) and their ability to predict responses to unseen images (y-axis). The black sideways triangle indicates the ceiling performance (median Spearman-Brown corrected split-half correlation across subjects and brain regions). Bars represent the mean response, with error bars showing the SEM across models and brain regions.

146 that have been reported extensively in prior work [14, 10, 9]. For each model, we focused on
147 features from an intermediate layer, selecting the layer that had previously been shown to achieve the
148 highest cross-validated accuracy in predicting responses from category-selective regions based on
149 an independent fMRI dataset [9]. This choice removed experimenter degrees of freedom. Next, we
150 constructed encoding models by mapping the features from a subset of images to brain responses
151 using a linear mapping function (see Methods for details on cross-validation procedures). This entire
152 process is depicted schematically in Figure 2A.

153 Overall, we found that these ANN-based encoding models were highly effective at predicting brain
154 responses to held-out images (replicating previous findings [10, 9]). This is illustrated for an example
155 brain region (EBA) in Figure 2B ($R = 0.76$, $P<0.00001$). Across all regions we considered, the
156 models were able to predict nearly all of the explainable variance in the observed data. The prediction
157 accuracy for each model architecture (Figure 2C, bars) was very close to the estimated noise ceiling
158 (Figure 2C, sideways triangle, derived from corrected split-half correlations). Importantly, all models
159 appeared to perform similarly well at predicting responses to unseen images. These results replicate
160 the earlier observation that a wide range of encoding models are approximately equal in their ability
161 to predict responses in the brain.

## 3.2 All ANN-based encoding models are susceptible to small scale adversarial attacks

163 How susceptible are encoding models to adversarial attacks? To address this, we engineered an
164 imperceptible noise pattern specifically designed to alter the predicted response for a given brain
165 region, along with a randomized noise pattern of the same magnitude and statistical properties as
166 a control. We discovered that even the slightest targeted noise, unseen by the human eye, could
167 completely derail the encoding model's predicted response. This is shown for an example encoding
168 model (VGG16) for the FFA and PPA in Figures 1C and 1D. Initially, the model's prediction for the
169 unaltered image from the preferred category (faces for FFA, scenes for PPA) was high. This agrees
170 with our expectation about images from the preferred category. However, adding a small amount of
171 targeted noise was enough to push the predicted response well outside the preferred category range
172 to the extreme end of the observed response spectrum. As a negative control, we used a shuffled
173 version of the same targeted noise. Importantly, this shuffled noise pattern, despite having the same
174 summary statistical properties of the noise, did not alter the predicted response to the same extent
175 (delta = 0.01).

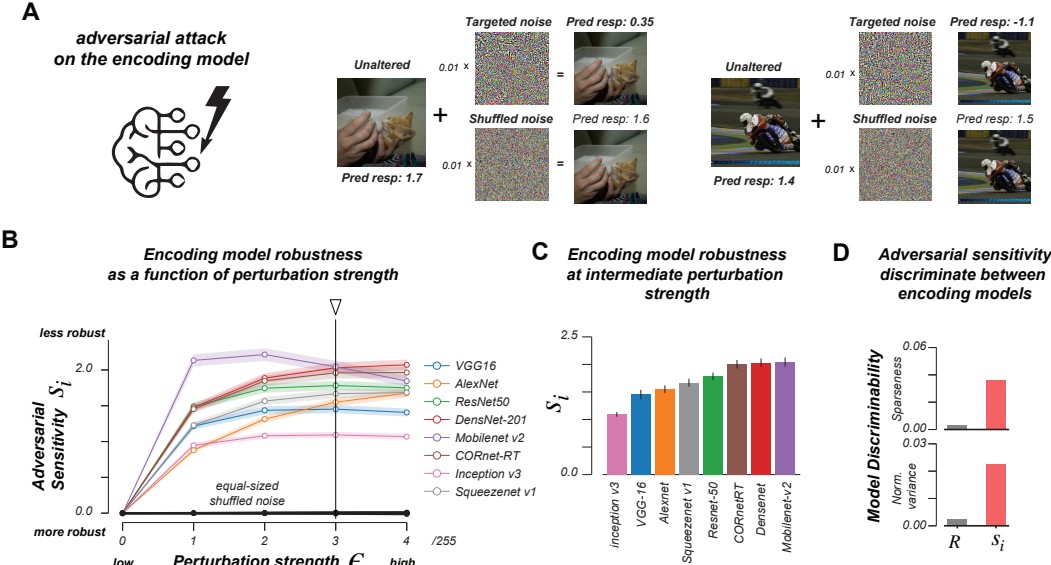

Figure 3: **Adversarial sensitivity effectively discriminates between encoding models. A.** Example of adversarial attacks applied directly to the encoding models. Each attack shows the unaltered image (left), the targeted noise and shuffled noise (middle panels top and bottom respectively), and model predictions for these images **B.** The effect of perturbation strength (x-axis) on the model's adversarial sensitivity (y-axis). Each colored line represents a different candidate encoding model architecture. The dots show the mean sensitivity, and the shaded areas represent the standard error across subjects and brain regions. The black line indicates the negative control using randomized noise. The triangle above marks the perturbation strength used for the subsequent analyses. **C.** Bar plots showing the adversarial sensitivity (y-axis) for all encoding models at a perturbation strength of $3/255$. The models are arranged in the same order as in Figure 2C for direct comparison. **D.** Barplots showing the discriminability between models using adversarial sensitivity and predictivity. Top: Bar plots illustrating model discriminability using predictivity ($R$) and adversarial sensitivity ($s_i$). Top: Discriminability based on the sparseness measure (y-axis). Bottom: Discriminability based on a normalized variance measure (y-axis).

We quantified the adversarial sensitivity for each model by measuring the change in predicted response to the adversarially perturbed image. Figure 3B shows these results for all encoding models. As the strength of the perturbation ($\epsilon$) increased (x-axis), the adversarial sensitivity also increased (as expected). Note that in this context, higher sensitivity indicates lower adversarial robustness for the model. These findings demonstrate that all tested ANN models were vulnerable to targeted adversarial attacks. In fact, for most models, even a small perturbation with $\epsilon = 3/255$ was enough to significantly alter the predicted response.

### 3.3 Adversarial sensitivity better discriminates between ANN-models than predictivity

Next, we evaluated whether adversarial sensitivity could serve as a more effective tool for distinguishing between candidate encoding models of the brain. We present these analyses for $\epsilon = 3/255$, although all subsequent inferences hold across other values as well. The results for adversarial sensitivity across all encoding models at $\epsilon = 3/255$ are displayed in Figure 3C. To facilitate comparison, the models are arranged in the same order as shown in Figure 2C.

To assess the effectiveness of adversarial sensitivity compared to predictivity, we employed two different measures. First, we measured the sparseness [61] of the adversarial sensitivity and predictivity metrics across models. Sparseness was chosen since it is a scale invariant measure and can be used to directly compare between predictivity and adversarial sensitivity (see Methods for details). Figure 3D (top) shows that model sparseness was significantly higher for adversarial sensitivity than for predictivity, indicating better discriminability across models. A problem with sparseness however is that it is highly sensitive to outliers. To allay this concern, we adopted a second, more

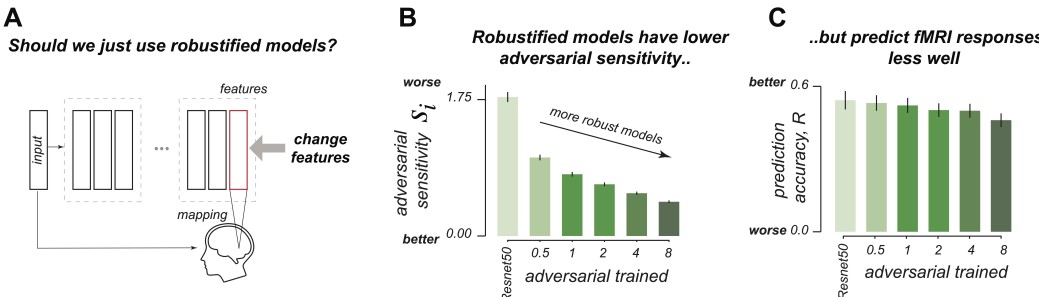

Figure 4: **Robust training reduces the predictive accuracy of fMRI encoding models. A.** Schematic illustration of the analysis. In this step, we replace the original features (shown in red) with features that have been robustified through adversarial training. **B.** Bar plots showing that increasing the level of adversarial training (x-axis) improves the adversarial sensitivity of the encoding models (y-axis). **C.** Bar plots showing that increasing the level of adversarial training (x-axis) reduces the predictive accuracy of encoding models on fMRI data (y-axis).

intuitive variance measure (normalized to match the scale between sensitivity and predictivity). As shown in Figure 3D (bottom), the normalized variance was also higher for adversarial sensitivity compared to predictivity. Together, these measures present a consistent picture: adversarial sensitivity distinguishes between encoding models more effectively than predictivity alone.

## 3.4 Increasing model robustness via adversarial training does not improve model predictivity

So far, we have demonstrated two key findings: 1) commonly used encoding models are sensitive to imperceptible adversarial noise, and 2) adversarial sensitivity can serve as a tool to distinguish between predictive models. How can we build better, more robust encoding models? The natural thing to try is to simply replace the current model architecture with a more robust one. In this section, we explored what happens when we use robustified models. To test this question, we fixed the model architecture (ResNet50) and parametrically varied the strength of adversarial training using publicly available robustified models [58]. This strategy is illustrated schematically in Figure 4A.

As expected, we found that robust models were indeed less vulnerable to added adversarial noise. Figure 4B shows how adversarial sensitivity decreases as the strength of adversarial training increases. Are robustified models effective at predicting fMRI responses? Here, we observed a trade-off: as the models became more robust, their ability to predict fMRI responses declined. This reduction was quite significant and is shown across all models and regions. These results suggest that while adversarial training does improves robustness, it may do so at the cost of reduced predictivity for brain data.

## 3.5 Sparse mappings tend to improve adversarial robustness of encoding models without sacrificing model predictivity

A less well-understood aspect of encoding models is the effect of the specific choice of mapping between model features and neural responses. We wondered if certain mapping functions could improve an encoding model's sensitivity to targeted noise. There are many potential linear and non-linear mapping functions to explore. To constrain our choices, we first evaluated five different mapping methods: ordinary least squares (no regularization), Lasso ($L_1$) regression (sparse), Ridge ($L_2$) regression, a two-layer multi-layer perceptron (MLP), and a convolutional neural network. We chose two candidate encoding models (VGG16 and ResNet50) for this initial exploration of mapping methods. An issue with these is that many of these methods involve choosing appropriate hyparameters. Hyperparameters were selected based on prediction accuracy (see Methods), but we focus our attention on an independent metric: adversarial sensitivity. The results are presented in Table 1. Across both models, we found evidence of a significant boost in adversarial sensitivity when using a sparse mapping.

Would this observation generalize to other models? To explore this, we compared the sensitivity of all eight models using $L_1$ (sparse) and OLS (no regularization) mapping-based encoding models across

**Adversarial sensitivity for different model-to-brain mapping functions**

| Model | OLS | **L1 (Lasso)** | L2 (Ridge) | 2-layer MLP | CNN |
|---|---|---|---|---|---|
| VGG16 | 1.453 | **.891** | 1.453 | 1.358 | 1.734 |
| ResNet50 | 1.782 | **.821** | 1.782 | 1.286 | 1.051 |

Table 1: Effect of readout functions on adversarial sensitivity. $L_1$ regularization on the readout performed best. The weight of the regularization term, $\alpha$, was chosen as the value which maximized predictive accuracy from a set of candidate values; see Methods.

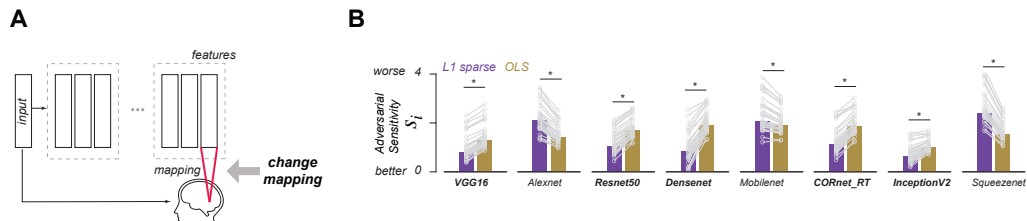

Figure 5: **Sparse model-to-brain mappings tend to lower adversarial sensitivity of encoding models . A.** Schematic illustration of the analysis. Here, we evaluate the model-to-brain mappings (highlighted in red) **B.** Bar plots showing adversarial sensitivity (y-axis) for all models tested. The dots and connected lines represent an encoding model for a specific subject and brain region. * indicates statistical significance (paired t-test, $P < 0.001$) between OLS and sparse mappings. Models in bold indicate improved adversarial sensitivity for sparse ($L_1$) mappings compared to OLS-based mappings.

all architectures. Note that the hyperparameters here were determined based on prediction accuracy from one subject, and the results were independently evaluated on adversarial sensitivity from all subjects (see Methods). This preliminary analysis revealed an interesting trend: sparse mappings produced significantly more robust models in 5 out of 8 model architectures. While this suggests that sparse mappings may enhance adversarial robustness, it is important to emphasize that these results are still preliminary and additional testing is needed to confirm whether this pattern holds across a larger sample, different model types, and independent analysis methods. Nonetheless, these early findings hint at the potential of sparse mappings to provide a meaningful boost in robustness.

## 4   Discussion

In this study, we investigated how susceptible commonly used ANN-based vision encoding models were to small-scale adversarial perturbations. We found that all high-performing models were vulnerable to imperceptible, small-scale adversarial noise (Figure 3). We also demonstrated that adversarial sensitivity, more effectively than prediction accuracy, could be used to differentiate between models (Figure 3). However, increasing model robustness through adversarial training came at the expense of reducing their ability to predict fMRI responses (Figure 4). Finally, we found early evidence that a simple sparse mapping approach on the mapping function could significantly improve adversarial robustness (Figure 5). These findings reveal key limitations of current encoding models and suggest new strategies for enhancing their performance.

Our adversarial attacks had two key features. First, the perturbations were deliberately kept small to focus on imperceptible changes. Our pilot analyses, based on an 8-degree viewing angle, suggest the detection threshold for adversarial images to be around $\epsilon = 8/255$. While a formal estimate on a larger sample is underway, we assumed that small perturbations, as those used in this study, would not alter brain voxel responses (though see [41]). This allowed us to test the model's vulnerability in a regime where the visual system should remain stable, highlighting its susceptibility to subtle adversarial noise. However, these assumptions require formal testing in future work. The second key feature is that our method targeted the encoding models directly (instead of the model features). This approach enabled us to assess vulnerabilities in the model's representational mappings to brain activity, not just the image embeddings. While previous studies have examined the relationship between model robustness and neural predictions in monkeys [44], or the link between spatial features

and neural representations [62, 63], our work extends these findings by exploring how adversarial perturbations *directly* affect model representations most predictive of human fMRI brain responses.

One interpretation of our results is that current high-performing, predictive encoding models are fundamentally flawed given how drastically they fail when exposed to targeted adversarial noise. While this is true, our aim is not merely to highlight these vulnerabilities. It is not entirely unexpected that these models are susceptible to adversarial perturbations, given what we know about neural networks in general. However, we propose leveraging adversarial sensitivity as a tool to guide the development of more accurate and resilient models. In fact, we find that adversarial sensitivity provides an additional layer of insight into model performance, helping to distinguish between highly predictive encoding models.

By analyzing how different models respond to adversarial perturbations, we start to uncover their limitations and use new insights into the development of more robust brain models. To this end, we tested two strategies. While adversarial training is widely used in the AI community to enhance model resilience, we found that it came at a significant cost to model predictivity (see also [44]). As models became more robust, their ability to accurately predict brain responses declined substantially. This trade-off highlights a compromise that must be carefully considered when developing models for neuroscience applications. In contrast, we found that a relatively simple sparse mapping between model features and brain representations was enough to significantly reduce the adversarial sensitivity of most encoding models, usually outperforming more complex non-linear mapping methods. We hope to explore these differences further in future work.

Taken together, our results expose the critical vulnerabilities of ANN-based predictive encoding models to adversarial perturbations, highlight adversarial sensitivity as a powerful tool for differentiating between models, and suggest a promising path for enhancing model robustness. As we continue our search for brain-like models, striking the right balance between robustness and predictivity will be crucial. Our work provides a foundation for tracking this balance, offers new model evaluations, and offers prescriptions to guide the development of more accurate and resilient models that can be applied to study human cognition even beyond vision.

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
