# OpenReview forum: "Small-scale adversarial perturbations expose differences between predictive encoding models of human fMRI responses"
_NeurIPS.cc/2024/Workshop/UniReps — UniReps_

### Official Review · Reviewer_PHKj · 2024-10-04
**Nice but needs more work**

**Rating:** 7
**Confidence:** 4

**Review:**

The paper discusses the observation that most encoding-decoding vision deep learning models achieve similar performance in explaining fMRI brain activations. To explore the differences between these models, the authors propose that robustness against adversarial perturbations, given the known human resilience to small adversarial visual attacks, could reveal the most "human-like" models. The study utilizes eight pre-trained models alongside two adversarially trained models to investigate this hypothesis.

The key findings indicate a significant difference in the models' sensitivity to adversarial perturbations, though the size of the perturbation appears to have limited impact (as shown by the early plateau in Figure 3C, which is somewhat unclear). The sensitivity to adversarial attacks seems to effectively distinguish between models (Figure 3D), although the estimation of these metrics is not fully explained.

The authors also observe that adversarial training reduces the models' sensitivity to attacks but comes at the cost of a reduction in their ability to predict fMRI responses. However, this finding is not entirely convincing—particularly since, although the models trained with adversarial examples do show significantly greater robustness according to Panel B, nevertheless the first three bars in Panel C of Figure 4 are not statistically significant based on the error bars.

Finally, the authors suggest that using sparser mappings to predict fMRI responses from the encoding models could improve robustness. However, the relevance of this suggestion is unclear. If the goal is to assess how adversarial perturbations affect model validity, it's not immediately obvious how changing the mapping contributes to this objective. The question of model alignment with fMRI responses seems related but distinct from the robustness of adversarial attacks.

In summary, the paper discusses a very important topic, suggests an intriguing approach to address it, but some of the interpretations are not well-supported by the presented evidence.

---

> ### Author Response · Authors · 2024-11-08
> **Response to reviewer PHKj**
>
> We thank the reviewer for their thoughtful and helpful comments.
>
> > The work uses multiple different brain regions for the analysis but only illustrates averaged one
>
> We did not find a significant difference between different models’ performances on brain regions, so we did not include figures with individual brain regions.
>
> > The key findings indicate a significant difference in the models' sensitivity to adversarial perturbations, though the size of the perturbation appears to have limited impact (as shown by the early plateau in Figure 3C, which is somewhat unclear).
>
> Yes, this an excellent point that the perturbation impact plateaus as the size of the attack increases. The reason for this is because we are using the Fast Gradient Sign Method (FGSM), which is a single-step linear attack (it finds the direction of the gradient which maximizes the loss and updates the image in a single step in that direction, bounding the total change by $\epsilon$). At some point, when the value of $\epsilon$ becomes too large, FGSM overshoots the optimal solution which maximizes the loss. When using a more sophisticated, nonlinear attack such as Projected Gradient Descent (i.e., computing the gradient at multiple steps before reaching epsilon), we do not find the early plateau. In fact, there is an approximately linear relationship between $\epsilon$ and $s_i$. The point that we would like to make is that, even with FGSM (which is a relatively low bound on how catastrophic an adversarial attack can be), all models are vulnerable, though they are each vulnerable to a different degree.
>
> > The authors also observe that adversarial training reduces the models' sensitivity to attacks but comes at the cost of a reduction in their ability to predict fMRI responses. However, this finding is not entirely convincing—particularly since, although the models trained with adversarial examples do show significantly greater robustness according to Panel B, nevertheless the first three bars in Panel C of Figure 4 are not statistically significant based on the error bars.
>
> We thank the reviewer for this important comment. In this section, we address the challenge of building models that are both predictive and robust to small-scale perturbations. Our results expose highlight an important key trade-off: adversarial training tends to reduce the models’ ability to predict fMRI responses. The reviewer is correct that the first three bars in Panel C of Figure 4 do not reach statistical significance based on the error bars. However, the overall trend is clear: blindly increasing robustness does not automatically improve a model’s ability to predict neural data. This trade-off underscores a deeper issue—achieving both high predictivity and robustness requires more than simple adversarial training. Our findings point to the need for more targeted strategies, such as selective adversarial training or hybrid approaches, to build models that balance robustness and neural predictivity. We have clarified this nuanced trade-off in the revised text.
>
> > Finally, the authors suggest that using sparser mappings to predict fMRI responses from the encoding models could improve robustness. However, the relevance of this suggestion is unclear. If the goal is to assess how adversarial perturbations affect model validity, it's not immediately obvious how changing the mapping contributes to this objective. The question of model alignment with fMRI responses seems related but distinct from the robustness of adversarial attacks.
>
> Thank you for the comment. To clarify, predictivity had previously been used a metric of model validity. Since most models now perform equally well with predictivity as a metric, our paper calls for a new metric to distinguish between models: predictivity + x. Here, the question about validity is now “prediction + adversarial robustness”. The relationship with the mapping function is that models become more valid with different mapping functions if they are all share the same level of predictivity. The point we would like to make is that the role of the mapping function is an under-explored area, and we believe that identifying the “right” mapping function to perform well on this problem is an interesting direction for future work.

---

### Official Review · Reviewer_f4xy · 2024-10-06
**Well-structured work that explores the impact of small-scale adversarial attacks on the brain encoding model's predictivity**

**Rating:** 7
**Confidence:** 4

**Review:**

**Summary:** The paper finds that encoding models are susceptible to imperceptible-to-human adversarial attacks and examines multiple models and mapping metrics using adversarial sensitivity.

**Quality, clarity and originality:** This paper is well-written and well-structured and the results fill gaps in the previous literature.

**Significance of the work:** This study is significant that it directly explores the relationship between adversarial sensitivity and brain prediction ability.

**Strengths:**
- This paper touches on lots of topics that interest the workshop audience
- The paper examines multiple different encoding models and mapping metrics

**Weaknesses:**
- The work uses multiple different brain regions for the analysis but only illustrates averaged one
- Recent findings suggest that CLIP is the new SOTA encoding model of the human brain (Conwell et al., 2023; Wang et al., 2023); it could be great to see CLIP included in the analysis too

---

> ### Author Response · Authors · 2024-11-08
> **Response to reviewer f4xy**
>
> We thank the reviewer for the encouraging comments.
>
> > The work uses multiple different brain regions for the analysis but only illustrates averaged one
>
> We did not find a significant difference between different models’ performances on brain regions, so we did not include figures with individual brain regions.
>
> > Recent findings suggest that CLIP is the new SOTA encoding model of the human brain (Conwell et al., 2023; Wang et al., 2023); it could be great to see CLIP included in the analysis too
>
> Thank you for the suggestion. Yes, we found that CLIP does perform well with predictivity but is also vulnerable to adversarial attacks. While this study focused only on models trained only on vision, we will showcase more models (including those trained with vision + language) in future work.

---

### Official Review · Reviewer_u3Kg · 2024-10-07
**Clever approach to differentiate equally predictive encoding models; hints at importance of sparse encodings**

**Rating:** 8
**Confidence:** 5

**Review:**

### Summary
This paper is extremely well-motivated, addressing the under-explored issue of adversarial robustness in brain encoding models. It has solid methodological novelty for a workshop paper, particularly in how the authors manipulate regression methods and explore predictivity and robustness across a variety of brain areas and models.

### Strengths
- **Novelty and Relevance**: We need better methods for differentiating encoding models, and looking at adversarial robustness is clever. I appreciate the authors' general philosophical stance toward the state of the field.
- **Clear Writing and Figures**: I was able to understand the analyses very easily.
- **Sparsity as a Key Insight**:  The connection between sparse encodings and improved robustness is really intriguing. Sparsity is also getting a lot of attention in explainable AI / interpretability, and this paper provides further evidence that there's something deep we should understand about the link between sparsity in models and brain function.

- **Findings on Adversarial Vulnerability**: It's not super surprising that linear encoding models are vulnerable to adversarial attack, but  it’s still valuable to show these signatures. It underscores the broader vulnerabilities of models used in cognitive neuroscience and helps make a case for further work in this space.

I think the paper is a clear accept for a workshop, however, there are a number of ways in which the work could be strengthened:

### Major comments
1. **Justification for Adversarial Robustness**: How strong is adversarial robustness as a litmus test for brain encoding models? If a model that could control neural populations at the voxel or neuron level were also susceptible to adversarial attacks, would that really make it a bad model?
2. **Proxy for Correct Features**: Relatedly, there could be stronger justification as to why adversarial sensitivity is a good proxy for whether a model has learned the “right” features. This feels like a bit of an indirect measure.
3. **Over-reliance on OLS Mapping**: OLS is prone to overfitting in my experience fitting encoding models of NSD. I wonder whether the trends reported in the first half of the paper would change if a method like Lasso or Ridge were used instead of OLS.
4. **Dataset Size**: The NSD dataset partitions (400/115) are probably fine for the analyses presented here, but it would be even better to train/tune hyperparameters on several thousand subject-specific images, and then test on the shared dataset of 515 images.  It is likely that prediction levels would plateau at train set sizes far larger than 400, especially for more flexible mapping methods.
5. **Cross-validation of Layers**: Selecting an intermediate layer based on prior work only seems acceptable if the methods, preprocessing, and fMRI data are exactly the same as that prior paper (Murty et al. 2021). Since they are not, the present study would benefit from cross-validating across layers rather than making an a priori selection.
6. **Adversarial Sensitivity Metric**: The metric seems to emphasize attacks that push activity down, but what about cases where low-response images suddenly drive activity? Any reason for lack of symmetry here?

### Minor Concerns
- **Alpha Value Sensitivity**: In my experience, small differences in the Lasso alpha parameter can have big effects on encoding outcomes. I suggest tailoring alpha values within subjects and looping the procedure over subjects, aggregating results at the end.
- **Voxels vs ROIs**: Predicting ROI means might lead to simpler, less informative solutions compared to fitting individual voxels. I wonder if the best model layers/regression schemes for fitting ROI-averages are more susceptible to attacks than the best layers/regression schemes for fitting groups of individual voxels. Fitting voxel-level data could help address this point.
- **Typo**: Line 225 contains a typo—“hyparameters” should be “hyperparameters.”

### Comments on Figures
- **Figure 2**:
   - The axes in panel B go from 0 to 8, but aren't responses typically in z-scored units for NSD?
   - In Fig. 2C, a risk of averaging over brain regions is that some models might perform much better in one ROI versus another. Is this the case?
- **Figure 3**:
   - Clarify how sensitivity is assessed over test images—is there just an averaging step? Also, can we interpret the units of sensitivity as a difference between pre- and post-attack brain activity predictions?
   - It would help to restate that OLS is used for regression in this context since the trends might be tied to OLS’s tendency to find dense solutions.
- **Figure 4**:
   - Missing details on the methods—what layer was used, and were different layers explored? If the same layer was chosen a priori, this raises questions about whether the most predictive layer simply shifts elsewhere in the model after adversarial training.
   - The decrease in prediction performance... is it tied to the drop in ImageNet recognition performance in robustified models? This should be presented clearly, with a table summarizing ImageNet performance across the models.
- **Figure 5**:
   - If Table 1 summarizes these results, that could be made more explicit.
   - While sparse encodings intuitively seem to defend against attacks, it’s confusing that OLS outperforms in 3 of 8 models. It suggests that adding a few more models might entirely flip the pattern of results. All of this raises even more questions about whether not cross-validating over layers and using the same alpha for all subjects might be impacting the overall trends.
   - The claim in Section 3.5 about not sacrificing predictivity with sparse mappings needs figure support—would be nice to show that L1 mappings are just as predictive as other methods.

### Discussion
- **Generalizability and Interpretation**: There’s a bit of ambiguity regarding what measuring adversarial sensitivity really tells us. While it’s interesting to point out vulnerabilities in these models, what new predictions or practical applications does this lead to? Is the claim that adversarial robustness is necessary for a model to be useful in brain encoding? Perhaps a bit of depth here would help the paper.
- **Sparsity**: There could be room to speculate in the Discussion: why might sparsity have a privileged role in these scenarios? Sparsity seems to be coming up a lot in neuro/AI... e.g. the use of sparse autoencoders for explainable AI, the idea that high-level features are sparsely represented in the data, the idea that information is routed through DNNs via sparse circuitry (e.g... https://arxiv.org/abs/2206.01627)
- **Other relevant citations to consider adding...**
   - Some work on using sparse DNN-brain encodings to improve model interpretability (see: https://openreview.net/forum?id=8FnN1QmR84).
   - There’s also work on failures of encoding models to handle distribution shifts (see: https://arxiv.org/abs/2406.16935v1).
   - And work on directly manipulating the sparsity of DNN representations to assess the impact on brain alignment: (see https://openreview.net/forum?id=ADDCErFzev).

Thanks for this timely and interesting work!

---

> ### Author Response · Authors · 2024-11-08
> **Response to reviewer u3Kg (1/3)**
>
> We thank the reviewer for their detailed and thoughtful comments.
>
>
> > Justification for Adversarial Robustness: How strong is adversarial robustness as a litmus test for brain encoding models? If a model that could control neural populations at the voxel or neuron level were also susceptible to adversarial attacks, would that really make it a bad model?
>
> The reviewer raises an excellent point that touches the core of our study. In the scientific context, “neural control” is a test of **precise experimenter-guided manipulation of neural responses** via a predictive computational model.
>
> A successful neural control experiment occurs when the experimenter modifies a stimulus to elicit an expected change of *X* and observes ***precisely*** that change. In our case, by applying a sufficiently small perturbation, we are essentially performing a neural control experiment. We modify the stimulus and the model predicts a neural response change of *X.* Note that the observed change should effectively be zero, since the perturbation lies below the resolution limit of the eye. The reveals a key insight: **the stimulus perturbation, while imperceptible to humans, impacts some models more than others.** Models that do not maintain their response essentially fail our version of the neural control test.
>
> > **Proxy for Correct Features**: Relatedly, there could be stronger justification as to why adversarial sensitivity is a good proxy for whether a model has learned the “right” features. This feels like a bit of an indirect measure.
>
> Related to the previous point, the core idea is that models which learn the “right” features, i.e. those that align with voxel representations, must predict held out data and also be robust to small, imperceptible perturbations.  If a model’s predictions break down under minor perturbations otherwise imperceptible to the human eye, it suggests that the model may be relying on spurious or fragile features rather than stable, generalizable ones that align with neural representations. We think adversarial sensitivity of encoding models might in fact be a stronger, direct test alongside predictivity measures.
>
> > **Over-reliance on OLS Mapping**: OLS is prone to overfitting in my experience fitting encoding models of NSD. I wonder whether the trends reported in the first half of the paper would change if a method like Lasso or Ridge were used instead of OLS.
>
> We have also tried Lasso and Ridge regression instead of OLS and did not find the predictive accuracy to be significantly higher on all models. We note that the test accuracy (on unseen data) is close to ceiling with OLS, suggesting that the model has not overfit the training data in our case.
>
> > Dataset Size: The NSD dataset partitions (400/115) are probably fine for the analyses presented here, but it would be even better to train/tune hyperparameters on several thousand subject-specific images, and then test on the shared dataset of 515 images. It is likely that prediction levels would plateau at train set sizes far larger than 400, especially for more flexible mapping methods.
>
> We appreciate the reviewer’s thoughtful suggestion. Since our analyses present results averaged across subjects, it was essential to ensure consistency across participants. The NSD dataset contains 515 shared images across all 8 subjects, which we used as a test set to evaluate generalization. Rather than looping over all of the data (a typical choice), we withheld 1/5th of the data entirely from the training pipeline, which we felt was a more stringent test of our models and highly consistent across subjects.
>
> And though the reviewer’s concern is valid, we confirmed that our analyses were robust to the number of training images. We find that the model prediction saturates for as less as 50 randomly chosen images (within correlation of 0.05 of max).

---

> > ### Author Response · Authors · 2024-11-08
> > **Response to reviewer u3Kg (2/3)**
> >
> > > **Cross-validation of Layers**: Selecting an intermediate layer based on prior work only seems acceptable if the methods, preprocessing, and fMRI data are exactly the same as that prior paper (Murty et al. 2021). Since they are not, the present study would benefit from cross-validating across layers rather than making an a priori selection.
> >
> > We appreciate the reviewer’s thoughtful suggestion and are happy to clarify our rationale. The dependence of layer selection on factors like methods, preprocessing, and fMRI data is in fact a testable hypothesis. In our preliminary analyses, however, we found that **the optimal layer primarily depends on the brain region, not on these methodological factors.**
> >
> > While it is straightforward to sweep across layers, doing so introduces additional experimenter degrees of freedom. A key strength of our approach is that it minimizes such flexibility—**our methodological pipeline is independent of subjective choices**, ensuring more reliable outcomes. We believe that selecting a layer based on prior work allows us to strike the balance between leveraging existing knowledge and maintaining methodological rigor. Following up on the reviewer’s comments we have also confirmed our analyses based on explicitly sweeping the model layers post-hoc.
> >
> > > Adversarial Sensitivity Metric: The metric seems to emphasize attacks that push activity down, but what about cases where low-response images suddenly drive activity? Any reason for lack of symmetry here?
> >
> > This is an excellent point. This is an explicit choice we made. We focused specifically on perturbations that caused models of certain brain regions to show low responses to objects from categories for which those regions are typically selective for interpretability. This choice in fact makes the observed sensitivity even more striking. The numbers here are a lower bound on the true sensitivity of these models. We will showcase the full range in future work.
> >
> > > Alpha Value Sensitivity: In my experience, small differences in the Lasso alpha parameter can have big effects on encoding outcomes. I suggest tailoring alpha values within subjects and looping the procedure over subjects, aggregating results at the end.
> >
> > We agree that tailoring the Lasso parameter for each subject could change the effects. In our experience, this only makes things better. As before, we wanted to be thorough and showcase the results without additional degrees of freedom.
> >
> > > Voxels vs ROIs: Predicting ROI means might lead to simpler, less informative solutions compared to fitting individual voxels. I wonder if the best model layers/regression schemes for fitting ROI-averages are more susceptible to attacks than the best layers/regression schemes for fitting groups of individual voxels. Fitting voxel-level data could help address this point.
> >
> > There is a tradeoff here. Voxels are more noisy than voxel-averaged responses. Also, the current hypotheses in cognitive neuroscience live at the level of ROIs, not individual voxels. Since this is our first report on adversarial sensitivity of encoding models, we chose interpretability over other factors. All of the core inferences hold (even more strongly) at the level of individual voxels. We will present this more comprehensively in future work.
> >
> > > Typo: Line 225 contains a typo—“hyparameters” should be “hyperparameters.”
> >
> > Oops. Fixed, thank you.
> >
> > > The axes in panel B go from 0 to 8, but aren't responses typically in z-scored units for NSD?
> >
> > The response units here are in percent signal change and not z-scored.
> >
> > > In Fig. 2C, a risk of averaging over brain regions is that some models might perform much better in one ROI versus another. Is this the case?
> >
> > While it is true that this is possible, we found that different models performed relatively similarly on different brain regions.

---

> ### Author Response · Authors · 2024-11-08
> **Response to reviewer u3Kg (3/3)**
>
> > Clarify how sensitivity is assessed over test images—is there just an averaging step? Also, can we interpret the units of sensitivity as a difference between pre- and post-attack brain activity predictions?
>
> Sensitivity is measured as the difference in the predicted response between unperturbed and perturbed images. We report the mean sensitivity across all test images for each model architecture. Yes, sensitivity reflects the difference between pre- and post-attack brain activity predictions.
>
> > Missing details on the methods—what layer was used, and were different layers explored? If the same layer was chosen a priori, this raises questions about whether the most predictive layer simply shifts elsewhere in the model after adversarial training.
>
> Yes, we report here using the same layers as in the previous analysis. While it is true that the most predictive layer could shift elsewhere, we did not find a significant difference in terms of predictivity across layers for different models.
>
> > The decrease in prediction performance... is it tied to the drop in ImageNet recognition performance in robustified models? This should be presented clearly, with a table summarizing ImageNet performance across the models.
>
> Yes, the ImageNet performance decreases as the level of robustness increases. These results have been previously reported [here](https://huggingface.co/madrylab/robust-imagenet-models).
>
> > If Table 1 summarizes these results, that could be made more explicit.
>
> Yes, the “Lasso” column in Table 1 summarizes the results for two of the models in Figure 5B. We have revised the manuscript to point this out.
>
> > While sparse encodings intuitively seem to defend against attacks, it’s confusing that OLS outperforms in 3 of 8 models. It suggests that adding a few more models might entirely flip the pattern of results. All of this raises even more questions about whether not cross-validating over layers and using the same alpha for all subjects might be impacting the overall trends.
>
> While it is true that OLS outperforms in 3 of the 8 models, the key point that we would like to make is that there is a difference in swapping the mapping functions. We agree with the reviewer that cross-validating over layers and subjects, as well as adding more models, will provide additional insight into the nature of the mapping function. This is an important direction for future work.
>
> > Other relevant citations
>
> Thank you! We have now included these important citations.

---

### Official Review · Reviewer_Rxq5 · 2024-10-07
**Distinguishing models using adversarial robustness**

**Rating:** 7
**Confidence:** 4

**Review:**

This paper looks at an important problem facing researchers attempting to compare artificial neural network models of the brain with neural data-- the fact that a wide variety of models seem to perform approximately equally in their predictivity of neural data.  This work considers the sensitivity of the model predictions of human fMRI data to adversarial noise patterns computed to maximally alter the predicted response of a model for a particular brain region.  They further show that using this analysis, adversarially trained models have lower adversarial sensitivity by are less predictive of fMRI responses.  They then briefly address how these results change when the mapping from model to brain data is changed, and suggest that a sparse mapping tends to reduce the adversarial sensitivity of the models.

The idea that different metrics besides neural data predictivity scores should be looked at to better distinguish artificial models of neural systems is certainly interesting.  However, I think the interpretation or utility of these results is not entirely clear.  The authors are able to use this approach to spread out these models in this choice of metric, but I don't know how to use this information to conclude anything about which deep networks are "better" models of the brain in some way (especially in light of the result about adversarially trained networks).  Should we conclude that the networks with higher robustness to these adversarial perturbations at human imperceptible levels are "better" models of the brain?  I would love to hear the author thoughts on this.  I think there is hope that further investigation along paths like this one will reveal more insights in this direction though, so I think this work is important.  The explanation of the mathematical machinery being used in this paper could definitely be improved, as the descriptions are vague and difficult to follow.  I would love to see more details in an appendix.

Some citations that may be missing:  I am reminded of the work by Feather et. al. Nat. Neuro (2023) that discusses using things like behavioral tests on synthetic stimuli (metamers and adversarial examples), and compares metamers behavior to encoding model performance.  Also, that of Harrington and Deza ICLR (2022).

Some questions and comments that arose during reading:

- for those unfamiliar with the Natural Scenes Dataset, what is tval?
- Paragraph starting at line 134:  Why do these regions provide the necessary foundational intuition for interpreting changes due to adversarial examples?
- Figure 1 C and D:  what exactly is being plotted on the y-axis here?
- Figure 2B:  does each dot represent a voxel?
- Figure 3A:  I'm not sure what the graphic on the top left is communicating
- I'm a little bit confused about the details of some of the computations in this paper, and I don't see an appendix.  For example, in section 2.3 when s_i is defined, I imagine it is a scalar representing a particular voxel.  However, earlier. g() is defined as a function that maps to vectors in R^n, where n is the dimensionality of the predicted neural data.  I think there is an index or perhaps a norm missing in this definition.  (minor thing: it would be great if you could label equations.)
- Continuing the previous point-- what exactly is being calculated when you say things like "change in predicted response?"  I thought the predicted response for each brain region would be a vector over voxels, however it is being summarized with a scalar in e.g. figure 3.  I understand the point that is being made but I think these details should be explained in the paper.  Mathematical definitions of what these quantities are would be helpful.

---

> ### Author Response · Authors · 2024-11-07
> **Response to reviewer Rxq5 (1/2)**
>
> We thank the reviewer for their thoughtful and encouraging feedback on our study.
>
> > Should we conclude that the networks with higher robustness to these adversarial perturbations at human imperceptible levels are "better" models of the brain?
>
> Our central argument is that adversarial sensitivity, especially at imperceptible levels, provides a powerful lens to examine the alignment of artificial networks with biological ones. To directly address the reviewer’s point: we do not claim that higher robustness to adversarial perturbations alone makes a model unequivocally “better”. Instead, we argue that **models which both predict neural data well and maintain robustness to subtle adversarial noise represent are better predictive models of neural responses**. The combination of predictivity and adversarial robustness might provide a richer, more nuanced measure for assessing models of neural responses.
>
> Importantly, we avoid framing our work in terms of “better models of the brain,” as we believe such language can be counterproductive. Our findings are more accurately summarized as identifying models that serve as **better predictive models of human visual responses**. This framing is more balanced, factually accurate, and grounded in measurable outcomes while leaving room for further exploration of biologically relevant features in future work.
>
> > The explanation of the mathematical machinery being used in this paper could definitely be improved
>
> Thank you for the comment. We have updated the mathematical notation in the paper to be more clear.
>
> > Some citations that may be missing
>
> Thank you! We have now included these important citations.
>
> > for those unfamiliar with the Natural Scenes Dataset, what is tval?
>
> We have explained this better now. Briefly, the NSD reports contrasts for 10 stimulus categories (Word, Number, Body, Limb, Adult, Child, Corridor, House, Car, instrument). The “facestval” for instance, is the contrast between faces (adult and child) and all other stimulus categories. This contrast is expressed as “tval”, the t-statistic from a two-sample t-test (for instance, faces versus other categories)
>
> > Paragraph starting at line 134: Why do these regions provide the necessary foundational intuition for interpreting changes due to adversarial examples?
>
> The extensive literature on category-selective regions offers strong priors on the expected response magnitudes. For example, the fusiform face area (FFA) is known to exhibit strong responses to faces and much weaker responses to non-preferred categories, such as scenes. This prior knowledge about its selectivity can be used to illustrate how adversarial noise can disrupt these predictable response patterns.
>
> As shown in Figure 1C and 1D show, imperceptible adversarial noise can reduce the response for an image from the preferred category (face) to  levels comparable with its non-preferred category (scene). Focusing on category-selective regions makes it easier to demonstrate the vulnerability of encoding models, which is why we prioritized these regions in out study. We have clarified this rationale more explicitly in the revised text.

---

> > ### Author Response · Authors · 2024-11-07
> > **Response to reviewer Rxq5 (2/2)**
> >
> > > Figure 1 C and D: what exactly is being plotted on the y-axis here?
> >
> > The y-axis is the observed response (for the 1000 shared images from NSD) as percent signal change (BOLD response). This figure legend has been edited for further clarity.
> >
> > > Figure 2B: does each dot represent a voxel?
> >
> > Each dot represents a given stimulus. We have updated the image caption to clarify this.
> >
> > > Figure 3A: I'm not sure what the graphic on the top left is communicating
> >
> > This graphic is merely an icon depicting the encoding model (brain+model combination, also shown in Figure 2A)  being “attacked” (the lightning).
> >
> > > I'm a little bit confused about the details of some of the computations in this paper, and I don't see an appendix. For example, in section 2.3 when s_i is defined, I imagine it is a scalar representing a particular voxel. However, earlier. g() is defined as a function that maps to vectors in R^n, where n is the dimensionality of the predicted neural data. I think there is an index or perhaps a norm missing in this definition. (minor thing: it would be great if you could label equations.)
> >
> > Thank you for catching this!  g(⋅) is a function which does return a vector (where the length is the dimensionality of the neural data predicted – in our case, voxels). Note that the adversarial attacks are computationally expensive, hence we show the *mean* over regions for a given subject and brain region (i.e., the mean of the vector returned by g(⋅). $s_i$ therefore refers to the average sensitivity of the *i-th* brain region/subject, not the *i*-th voxel. We have revised the manuscript to clarify this.
> >
> > > Continuing the previous point-- what exactly is being calculated when you say things like "change in predicted response?" I thought the predicted response for each brain region would be a vector over voxels, however it is being summarized with a scalar in e.g. figure 3. I understand the point that is being made but I think these details should be explained in the paper. Mathematical definitions of what these quantities are would be helpful.
> >
> > The “change in predicted response” refers to the average $s_i$ per model. In other words, for a given subject and brain region, we obtain clean responses to 115 images and then responses to 115 perturbed images. We calculate the average difference, and this is $s_i$. Then, we average over i brain regions and subjects, yielding our estimate for the model change in response. We have also updated the manuscript to reflect this.

---

### Decision · Program_Chairs · 2024-10-10

**Decision:**

Accept

**Comment:**

In light of the positive reviewers' feedback and relevancy of the submission, we are pleased to accept this paper for presentation at UniReps 2024. We kindly ask the authors to incorporate the reviewers' suggestions and feedback in the final camera-ready version of the manuscript.